# LISA: UNLEASHING 2D DIFFUSION FOR 3D GENERATION VIA LIGHTWEIGHT IMAGE SPLATS ADAPTATION

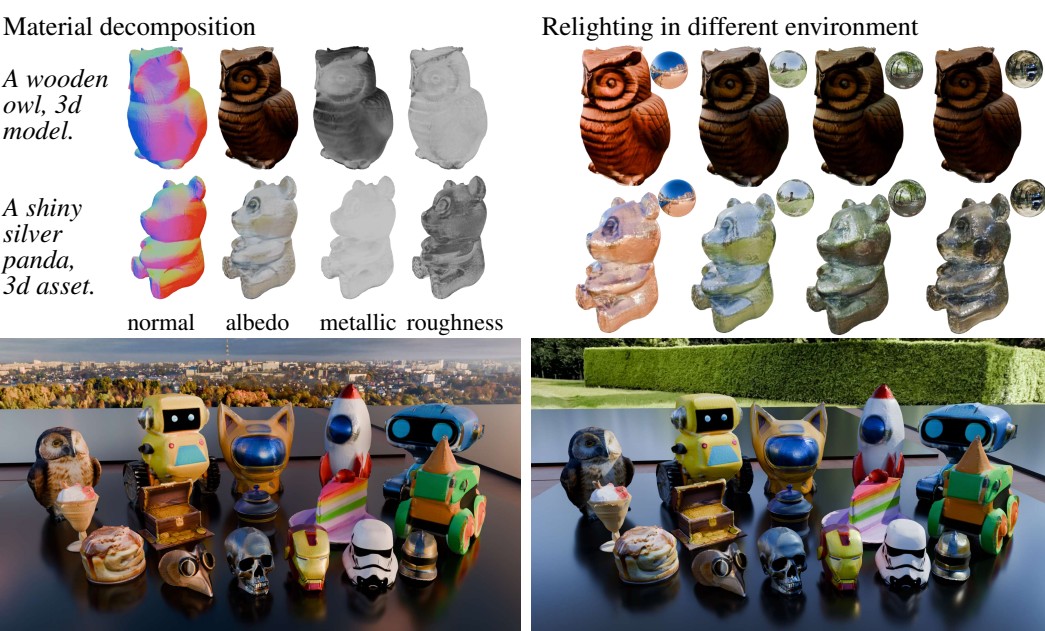

Figure 1: Given text prompts, **LISA** can generate high-quality meshes with PBR materials on a single RTX 3090 GPU within 30 seconds.

## ABSTRACT

Despite its potential, 3D generation lags behind 2D generation in quality and utility, primarily due to the vast gap in the scale and diversity of training data—high-quality 2D data is abundant, while high-quality 3D assets remain limited by orders of magnitude. Existing methods use 2D generative priors for 3D asset creation via distillation or generate-and-reconstruct schemes, both of which suffer from quality loss during optimization. In this paper, we propose a novel scheme to exploit 2D diffusion prior to 3d generation by integrating a lightweight adapter into the decoder of a frozen 2D diffusion model, allowing it to generate RGB images, Gaussian splats, and physics-based rendering material maps simultaneously. Once trained, the proposed **L**ightweight **I**mage **S**plats **A**daptation (**LISA**) directly produces relightable Gaussian splats in feed-forward manner, which can be converted into high-quality, relightable 3D meshes through an inverse rendering framework. Quantitative and qualitative results demonstrate that our method outperforms state-of-the-art approaches with a significantly lower computational budget for both training and sampling. More results can be found at https://LISA-3dgen.github.io.

## 1 INTRODUCTION

AI-generated content(AIGC) has achieved remarkable progress in recent years, especially in 2D image generation Rombach et al. (2022). However, 3D generation has been left far behind with 3D

datasets of limited scale and quality. Though suffering from time-consuming per-prompt optimization and 3D inconsistency, Dreamfusion Poole et al. (2022) and its follow-up works Wang et al. (2023); Chen et al. (2023); Lin et al. (2023); Wang et al. (2024a); Chen et al. (2024b); Tang et al. (2023) achieve zero-shot 3D generation via score distillation from pre-trained 2D diffusion priors. To better leverage 2D priors, pioneer works Shi et al. (2023a); Long et al. (2024); Liu et al. (2023a); Shi et al. (2023b) fine-tune stable diffusion models on the large-scale open-world 3D dataset Deitke et al. (2023; 2024); Yu et al. (2023a) to inject 3D consistency prior into 2D diffusion models, known as multi-view(MV) diffusion models. Large reconstruction models(LRMs) Li et al. (2023a); Hong et al. (2023); Xu et al. (2023b; 2024a); Tang et al. (2024); Xu et al. (2024b) lift MV images to 3D representation Mildenhall et al. (2021); Wang et al. (2021a); Shen et al. (2023); Kerbl et al. (2023) via a large feed-forward model. Although the latter diagram reconstructs 3D representation from multi-view images, the LRMs are trained on 3D datasets with limited scale and have no direct access to 2D diffusion priors, failing to explore 2D diffusion priors encoded in the network fully.

*Does a 2D diffusion model already encode 3D information?* We conduct a toy experiment to investigate this. We take an off-the-shelf multi-view latent diffusion model and attach a lightweight decoder, allowing it to output a splatter image Szymanowicz et al. (2024) (where each pixel has a Gaussian splat's mean, opacity, and covariance features instead of RGB color). We only fine-tune this decoder on a toy dataset that contains multiple 3D plants, while freezing all other learnable components. Surprisingly, we find that this new model is able to generate complete 3D Gaussian assets, even for input prompts that are not plants, such as animals and robots. This suggests that (1) multi-view latent diffusion might already encode 3D knowledge; we just need a lightweight adaptation to unleash it, allowing it to output 3D, and (2) such 3D generation can retain the rich knowledge learned in 2D diffusion and transfer it to 3D without forgetting, thereby enabling us to leverage widely available 2D data.

Inspired by the findings above, we present the **L**ightweight **I**mage **S**plats **A**daptation (**LISA**) to repurpose 2D diffusion for end-to-end realistic, relightable, and generalizable 3D asset generation, as shown in Figure 1. LISA achieves efficient end-to-end 3D generation by directly adapting pre-trained layers from 2D diffusion models to output Gaussian splat images via additional learnable zero convolution layers. We can easily fuse the Gaussian splats from multiple views and ensemble a complete 3D Gaussian asset. This lightweight adaptation approach maximizes the preservation of rich generative priors learned from 2D data while generalizing for 3D creation. Moreover, it benefits from low 3D training data requirements as well as fast sampling and rendering speeds. To produce realistic and user-ready 3D assets, we further apply test-time inverse rendering to convert our 3D Gaussians into high-quality, UV-mapped, relightable 3D meshes.

Our experiments show that LISA achieves superior performance over prior 3D generation approaches in terms of both geometry quality and rendering quality, demonstrating the efficacy of the proposed framework. Notably, LISA achieves this with significantly less 3D training data and faster generation times. Specifically, with a subset of 46K high-quality multi-view instances from G-Objaverse Qiu et al. (2024), LISA can be efficiently fine-tuned to generate 2D Gaussian splats with coarse PBR materials, based on which we further achieve high-quality mesh generation with PBR materials in under 30 seconds on a single RTX 3090 GPU.

Our contributions are as follows:

- We find that 2D diffusion networks can be adapted to directly generate 3D Gaussian splats using a lightweight decoder with data- and compute-efficient training, and present the **LISA** decoder to exploit 2D diffusion models for direct 3D generation.
- We present a novel text-to-3D generation scheme that efficiently generates high-quality, relightable, and realistic 3D assets with consumer-grade GPUs.

## 2 RELATED WORKS

**Text-to-3D with 2D diffusion priors.** With the development of diffusion theory Ho et al. (2020) and the emergence of various neural 3D representations Mildenhall et al. (2021); Wang et al. (2021a); Shen et al. (2023); Kerbl et al. (2023); Shen et al. (2021); Mescheder et al. (2019), text-to-3D has achieved significant progress in terms of quality and speed. Dreamfusion Poole et al. (2022) propose Score Distillation Sampling(SDS) loss to distill 3D consistent NeRF from text-to-image(T2I)

diffusion priors given only text prompts, which open up a new era for zero-shot 3D generation and follow-up efforts that improve distillation quality Yu et al. (2023b); Liang et al. (2024); Wang et al. (2024a), apply on different neural representations Chen et al. (2023; 2024b); Lin et al. (2023); Yi et al. (2023); Tang et al. (2023), text-to-3D scene generation Zhang et al. (2024); Fang et al. (2023), and even extend to 4D generations Ren et al. (2023); Singer et al. (2023); Ling et al. (2024) occur like mushrooms after rain. However, due to the lack of 3D priors in T2I diffusion models, distillation-based methods suffer from the 3D inconsistency problem, also known as the Janus problem; therefore, the community paves the way to inject multi-view 3D priors into T2I models by fine-tuning pre-trained models using rendered views from large-scale dataset Deitke et al. (2023; 2024) to generate multi-view consistency images Shi et al. (2023b); Liu et al. (2023a); Long et al. (2024); Shi et al. (2023a); Qiu et al. (2024); Li et al. (2023b), which server as strong 2D and 3D combined priors and improve the quality of generated 3D assets by a large margin.

**Multi-view 2D diffusion priors for 3D generation.** As distillation-based methods still suffer from time-consuming per-prompt optimization, instant3D Li et al. (2023a) propose the diagram to decompose the text-to-3D generation task into text-to-MV images generation and MV-to-3D generation, the former phase is implemented as a fine-tuned multi-view 2D diffusion model, while the latter phase features a feed-forward network mapping multi-view images to NeRF representation. The two-stage diagram indirectly benefits from both 2D and 3D priors and demonstrates superiority against previous methods regarding quality and speed. InstantMesh Xu et al. (2024a) and CRM Wang et al. (2024b) extend the diagram to direct mesh generation, while LGM Tang et al. (2024) and GRM Xu et al. (2024b) build the reconstruction models with 3D Gaussian Splatting. However, such models are all trained from scratch and fail to directly reuse 2D priors. Therefore, the training process consumes tens to hundreds of high-end GPUs, which is typically unaffordable in the academic community. Concurrent work LaRa Chen et al. (2024a) proposes to leverage the pre-trained 2D feature encoder Caron et al. (2021) to construct dense 3D volumes and regress 2D Gaussian primitives from them. Though LaRa achieves remarkable results on a limited budget, 3D volume-based regression networks are GPU-memory-intensive designs, which will limit the scalability and downstream applications.

**3D generation with PBR materials.** Simultaneously recovering geometry, materials, and illuminations is a highly ill-posed problem even from densely captured data Liu et al. (2023b); Zhu et al. (2024); Zhang et al. (2021); therefore, most existing works generate meshes with simple baked colors, which are not compatible with modern graphics pipeline. Liu et al. (2023c); Xu et al. (2023a); Qiu et al. (2024) introduce PBR priors into the optimization process to achieve material decomposition during the generation procedure, while Siddiqui et al. (2024) propose a feed-forward network to regress SDF fields with coarse PBR materials and refine the textures by a texture refiner network.

## 3 METHODS

Our proposed pipeline is shown in Figure 2, based on which we seamlessly adapt 2D diffusion models for direct high-quality 3D asset generation. Firstly, we briefly revise the related 3D representation and diffusion models in Section 3.1. Then, we introduce the building blocks for our LISA model by experimentally adapting MVDream for 3D Gaussian splats generation in Section 3.2, which also demonstrates the benefits of reusing 2D diffusion priors. Next, we describe the details of our LISA model in Section 3.3. To improve the usability of generated 3D assets, we design an automatic post-processing procedure to convert our generated Gaussian splats into high-quality meshes with PBR materials in Section 3.5.

### 3.1 PRELIMINARIES

**2D Gaussian Splatting.** Kerbl et al. (2023) propose 3D Gaussian Splatting (3DGS) to parameterize the 3D scene via radiance fields in the form of a collection of Gaussian primitives $\mathcal{G} = \{g_i\}$, where each primitive contains multiple attributes recovered by differentiable rendering. However, 3DGS fails to recover accurate geometry surface. Therefore, Huang et al. (2024) improve the representation by simplifying each primitive into 2D Gaussian Splatting (2DGS), each of which is parameterized by a 3D position $\mu \in \mathbb{R}^3$, a rotation vector $\mathbf{R} \in \mathbb{R}^3$, a scaling vector $\mathbf{S} \in \mathbb{R}^2$, an opacity $o \in \mathbb{R}$, and a view-dependent appearance $\mathbf{c} \in \mathbb{R}^{(d+1)^2 \times 3}$ represented by spherical harmonic of degree

(a) LISA model (Sec. 3.2) (b) Relightable GS generation (Sec. 3.3) (c) Post processing (Sec. 3.5)

Figure 2: Overview of the proposed pipeline. We decompose the generation process into three parts. Firstly, we generate 2DGS with PBR materials using our LISA model. Then, we extract mesh via TSDF Fusion, perform continuous remeshing to refine geometry, unwrap the uv map for the refined mesh, and initialize the PBR texture maps by projecting rendered views from 2DGS onto the mesh. Finally, we align the rendered views of the mesh with generated 2D views from MVDream to refine the PBR materials via differentiable ray-racer Jakob et al. (2022b).

*d.* As an explicit representation, 2DGS is an unstructured represetation, which is incompatible with traditional 2D neural networks. Inspired by Splatter Image Szymanowicz et al. (2024), we leverage 2DGS and organize them in the form of multi-view Gaussian attribute maps, where each pixel represents one Gaussian primitive, therefore, we can easily leverage 2D neural networks to generate 2DGS while guarantee accurate geometry extraction.

**Multi-view diffusion model.** Multi-view(MV) diffusions are typically fine-tuned from stable diffusion Rombach et al. (2022) to generate 3D consistent MV images. MVDream Shi et al. (2023b) is fine-tuned to generate four orthogonal views around the object with elevation at the range of $[0, 30]$ through the denoising process. MV normal depth diffusion Qiu et al. (2024) is trained to directly generate MV normal and depth images. The above diffusion priors contain strong 2D and 3D priors.

## 3.2 LIGHTWEIGHT IMAGE SPLATS ADAPTATION

Except for current ways to leverage 2D diffusion priors for 3D generation via distillation or generate-and-reconstruct scheme, we opt to exploit another way to achieve end-to-end 3D generation by modifying multi-view diffusion models to output 3D representation. However, the challenge mainly lies in the entirely different organization of primitive data. To mitigate the gap, we construct the generation part using the Gaussian Splatting organized in multi-view Gaussian attribute maps as mentioned in Section 3.1, and for simplicity, we use 3DGS instead of 2DGS in this section.

With proper 3D representation, we propose to extract 3D information from 2D diffusion models without violating pre-trained priors. ControlNet Zhang et al. (2023) paves an efficient way to inject control information into a large pre-rained diffusion model via a new branch adapted by zero convolution layers, inspired by which we present our **L**ightweight **I**mage **S**plats **A**daptation (**LISA**) to achieve direct 3D generation, except that ControlNet is designed to modulate extra information into the diffusion model while LISA is to demodulate extra information from the model.

We introduce our LISA model from the basic building block, which we name the LISA block and LISA switcher, as in Figure 3. For simplicity, we follow the notation from Zhang et al. (2023), and refer *network block* in diffusion models as commonly combined network blocks, such as resnet block, conv-bn-relu block, transformer block, *et al*. The LISA block clones and freezes the pre-trained layer from 2D diffusion models, adapts the information flow from the previous layer, skip connection, and intermediate information from frozen 2D diffusion models with zero convolution layers. With such a design, the LISA block outputs the same information with the same inputs given at the start of training. Therefore, the constructed model maximizes the preservation of rich generative priors learned from 2D data while generalizing for 3D creation. As we unleash 2D diffusion priors to output extra information as 3D representation, we construct LISA with the decoder part of the 2D diffusion models.

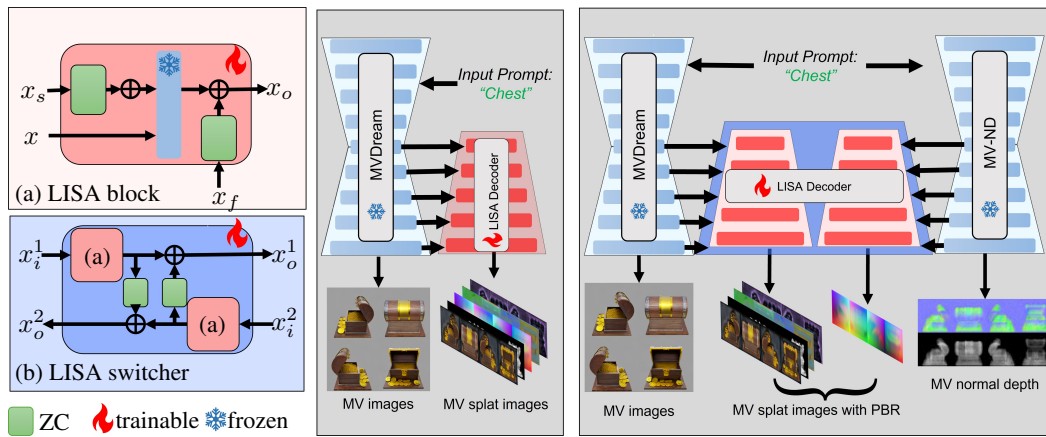

Figure 3: Illustration of LISA constructed with different 2D diffusion priors. ZC stands for zero convolution, which is used to adapt the frozen layers from 2D priors. GS represents output heads for GS attribute maps. In (a), $x$ represents the feature map from the former layer, $x_s$ is the one from the skip connection, $x_f$ is intermediate information from 2D diffusion models, and $x_o$ stands for output. Block (b) is based on (a) and used for information switching between different 2D diffusion priors, where $x_i^1$ and $x_i^2$ represent feature maps from the former layer in their branch while $x_o^1$ and $x_o^2$ stand for output feature maps. Due to the flexibility of our block, we can construct a LISA model with one 2D diffusion prior, as shown in (c) with the LISA block, or fuse multiple 2D priors to a LISA model, as in (d) with additional switcher blocks.

To verify our design, we build a simple LISA model with MVDream Shi et al. (2023b) being the 2d priors, which is fine-tuned from stable diffusion to generate four views with orthogonal azimuth angles and an elevation angle in the range of $[0, 30]$, as shown in Figure 3(c). We train the model on a subset of multi-view images labeled "Plants" from G-Objaverse Qiu et al. (2024), which we provide details in Section 4.1. During training, we follow Shi et al. (2023b) to generate the required inputs for multi-view models and extract the needed intermediate information

$$\{\mathbf{f}_i\} = \mathcal{MV}(\mathbf{x}_t; y, \mathbf{c}, t) \tag{1}$$

where $\mathcal{MV}$ is the pre-trained MVDream, $t$ is the noise level, $\mathbf{c}$ is the camera pose, and $\mathbf{x}_t$ is the noisy orthogonal views with random noise $\epsilon$ at noise level $t$. With intermediate information $\{\mathbf{f}_i\}$, we then generate the Gaussian attribute map as

$$\mathcal{G} = \mathcal{F}(\{\mathbf{f}_i\}; y, \mathbf{c}, t) \tag{2}$$

where $\mathcal{F}(\cdot)$ is our generation network, $\mathcal{G}$ is the predicted Gaussian attribute maps at the shape of $[4 \times d \times H \times W]$, $d$ is the attribute channel for each Gaussian primitive, and $H = W = 32$ is the height and width for feature maps. The generated Gaussian attribute map $\mathcal{G}$ is at a very low resolution, resulting in a very sparse 3DGS with just 4096 primitives, and the representation ability is severely limited. Therefore, we initialize 16 embedding vectors and repeatedly add the embedding vectors to the feature maps before Gaussian attribute heads to decode 16 pairs of Gaussian attribute maps $\{\mathcal{G}_i\}_{i=1}^{16}$, which forms a dense 3DGS with 65,536 primitives.

We supervise the model by rendering eight views from the 3DGS, consisting of the four orthogonal views input to MVDream and four random views, and we utilize MSE loss and SSIM loss at the resolution of 256 along with LPIPS loss at the resolution of 128. During inference, we run DDIM Song et al. (2020) sampling for MVDream, and when the noise level is lower than 80, we feed the intermediate information from MVDream into our GS prediction model once to generate the corresponding 3DGS representation. As shown in Figure 4, given prompts out of the training domain, our model learns to adapt the knowledge from MV image generation into 3DGS generation. Note that the model is only trained on two NVIDIA RTX 3090 GPUs.

### 3.3 LISA WITH MULTIPLE PRIORS

Our toy experiment in Section 3.2 demonstrates the possibility of reusing 2D diffusion models for direct 3D generation via our LISA model. Tough directly reusing 2D priors for 3D generation shows

Figure 4: We fine-tune our model on the shown plant-shaped multi-view data, and the model demonstrates strong generalization ability to generate out-of-domain instances, while the generated 3DGS are consistent with 2D image priors.

great potential, the results shown in Figure 4(b) still suffer from blurry results, which is mainly due to the low resolution of the Gaussian attribute map and the placement of each Gaussian primitive needs more accurate 3D priors instead of just multi-view priors. To improve the generation quality, we propose to combine multiple 2D diffusion priors for better Gaussian primitive placement and upsample the Gaussian attribute maps to improve reconstructed details.

As shown in Figure 3(d), we design parallel LISA branches to leverage multiple priors. To align different diffusion priors for single 3D asset generation, we inject information between each branch via a LISA switcher module, which densely connects parallel LISA branches via zero convolution layers, allowing the gradual construction of an information bridge. Since the Gaussian attribute maps are separated into different independent maps, we can flexibly decode the attributes using different parts of our network. Therefore, we assign the position generation task to MV normal depth diffusion Qiu et al. (2024) as it encodes priors related to geometry, and we generate other attributes using the MVDream diffusion prior to model the appearance of the 3D instance. Further, we leverage pre-trained a lightweight super-resolution model Wang et al. (2021b) to upsample the feature maps by a factor of 4 to decode Gaussian attribute maps at a resolution of 128, and we lock the pre-trained super resolution block and add trainable layers before and after the block, which helps stabilize the training progress while builds connections between each primitive in the feature maps.

To improve the usability of the generated 3D assets, we employ 2D Gaussian Splatting Huang et al. (2024) as our representation, and we add heads to decode albedo, metallic, and roughness. Besides, we project the Gaussian primitives into 3D scenes via depth and offset to guarantee more accurate placement. Specifically, each primitive in the generated Gaussian attribute maps consisting of a depth $t \in \mathbb{R}$, a 2D offset $\mathbf{p} \in \mathbb{R}^2$, a rotation vector $\mathbf{R} \in \mathbb{R}^3$, a scaling vector $\mathbf{S} \in \mathbb{R}^2$, an opacity $o \in \mathbb{R}$, a metallic value $m \in \mathbb{R}$, a roughness value $r \in \mathbb{R}$, and a view-dependent appearance $\mathbf{c} \in \mathbb{R}^{(d+1)^2 \times 3}$ represented by spherical harmonic(SH) of degree $d = 2$. For the view-dependent appearance $\mathbf{c}$ we predict the DC component via the albedo head, and leave the other channels to the SH head. We project each primitive into the 3D world via $\mu = \mathbf{o} + t \cdot \mathbf{d}$, where $\mathbf{o} \in \mathbb{R}^3$ is the camera center and $\mathbf{d} \in \mathbb{R}^3$ is the view direction, which is calculated after we move the primitive with offset $\mathbf{p}$ in the NDC space. As MVDream tends to generate multi-view images with random background colors, we use an additional head to decompose a background color $\mathbf{c}_b \in \mathbb{R}^3$ from the feature map before upsampled. Also, we preserve the original denoising head for our network to stabilize the model.

### 3.4 TRAINING AND INFERENCE

**Training.** During training, we sample a batch with four orthogonal views and four random views, each of which consists of RGB, alpha mask, normal, depth, albedo, metallic, and roughness at the resolution of 256. We train our model using bfloat16 precision and gradient checkpointing steps of 16, with each GPU processing one batch, resulting in a total batch size of 128. We add random grid distortion Tang et al. (2024) to the four orthogonal views, apply a random background color to the RGB images, and then process the views following MVDream and MV-normal-depth with random noise level $t \in [0, 1000]$ to get the input for the model. Then, we render the eight views of RGB, albedo, alpha map, metallic, roughness, depth, and normal at the resolution of 256. For RGB and

albedo supervision, we apply MSE loss, SSIM loss, and LPIPS loss:

$$\mathcal{L}_{color} = \lambda_1 \mathcal{L}_{MSE}(\mathbf{I}_{color}, \mathbf{I}_{GT}) + \lambda_2 \mathcal{L}_{SSIM}(\mathbf{I}_{color}, \mathbf{I}_{GT}) + \lambda_3 \mathcal{L}_{LPIPS}(\mathbf{I}_{color}, \mathbf{I}_{GT}) \quad (3)$$

where $\lambda_1 = 1, \lambda_2 = 2, \lambda_3 = 5$. For alpha map, we use binary cross entropy loss

$$\mathcal{L}_{alpha} = (\lambda_1 + \lambda_2 + \lambda_3)\mathcal{L}_{BCE}(\mathbf{I}_{alpha}, \mathbf{I}_{GT}) \quad (4)$$

For metallic and roughness, we apply MSE loss

$$\mathcal{L}_{material} = (\lambda_1 + \lambda_2 + \lambda_3)\mathcal{L}_{MSE}(\mathbf{I}_{material}, \mathbf{I}_{GT}) \quad (5)$$

And we also apply the depth distortion loss with a weight of $2e4$ and normal consistency loss from Huang et al. (2024).

**Inference.** We infer our model with DDIM sampling with 50 steps and a guidance scale of 7.5 for the MVDream and MV normal-depth model. As our model directly decodes multi-view splat images from multi-view diffusion models, we only take the generated multi-view splat images generated when the noise level is lower than 80. To align the MV normal depth model with MVDream, we perform sampling using the denoising head in LISA decoder instead of the original one in the MV normal depth model.

### 3.5 GEOMETRY AND TEXTURE REFINEMENT

Based on the generated 2DGS from our feed-forward network, we first extract meshes from 2DGS via TSDF Fusion and refine them through continuous remeshing Palfinger (2022), then initialize texture maps for the 3D assets and leverage differentiable ray-tracer to refine the PBR materials.

**Geometry extraction and refinement.** To extract meshes from 2DGS, we render albedo and depth along circle camera paths at elevations of $[10, 15, 20]$ around the instance plus a top view and a bottom view, and we use the ScalableTSDFVolume from open3d Zhou et al. (2018) with the voxel size of 0.004 and the truncation threshold of 0.02 to perform TSDF Fusion to extract the initial mesh. Then, we extract the convex hull of the initial mesh to fill all the holes in the original mesh. Finally, we render normal maps and alpha maps from the 2DGS along circle paths at elevations of $[-40, -30, -20, -10, 0, 10, 20, 30, 40]$ around the instance as the target views and perform 100 iterations of continuous remeshing Palfinger (2022) to transform the convex hull into high-quality, smooth meshes.

**Texture initialization and refinement.** After mesh extraction, we use the smart uv project from Blender Community (2018) to generate UV maps for the mesh, and we unproject the rendered albedo, metallic, and roughness from 2DGS onto the mesh to initialize the PBR materials. Then following Ummenhofer et al. (2024), we use differentiable ray-tracer from Jakob et al. (2022b;a) to align the four orthogonal images with the generated four views from MVDream.

## 4 EXPERIMENTS

We introduce the details of the dataset and the training settings in Section 4.1, then in Section 4.2 we report quantitative comparisons with MVDream via FID Heusel et al. (2017), Inception score(IS) Salimans et al. (2016), and CLIP score Radford et al. (2021), and provide the user study with LGM Tang et al. (2024) and LaRa Chen et al. (2024a), as which are Gaussian-based generation methods. Finally, we exhibit qualitative comparisons between our methods with LGM, LaRa, Meshy-3, and LumaAI Genie.

### 4.1 DATASET AND IMPLEMENTATION DETAILS

**Dataset.** We use a subset of G-Objaverse Qiu et al. (2024) as our training data, which is a large scale multi-view dataset rendered from Objaverse Deitke et al. (2023). G-Objaverse renders 264,775 instances in total, which renders 38 views for each instance, consisting of 24 views at an elevation range of $[5, 30]$ and evenly distributed around the object along the azimuth channel, 12 views at an elevation range of $[-5, 5]$ and evenly distributed around the object along the azimuth channel, plus one top view and one bottom view. For each view, it provides rendered RGB, albedo, metallic,

*Star Wars Stormtrooper helmet, single object.*

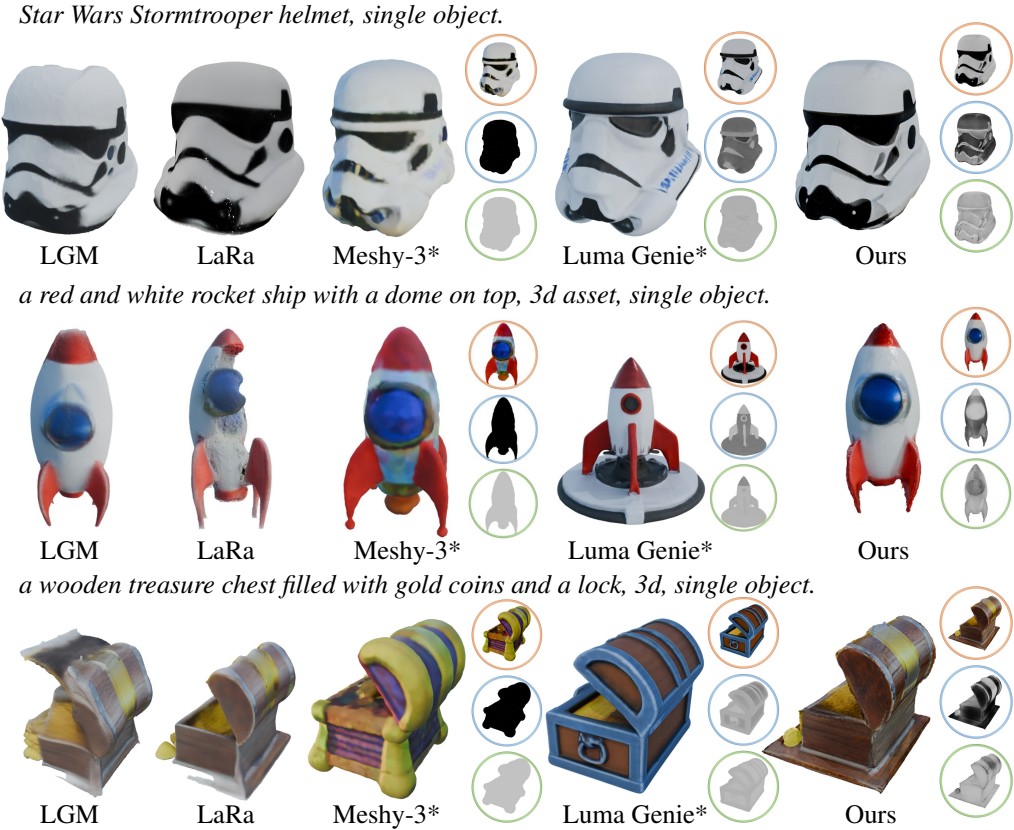

LGM    LaRa   Meshy-3*   Luma Genie*   Ours

*a red and white rocket ship with a dome on top, 3d asset, single object.*

LGM    LaRa   Meshy-3*   Luma Genie*   Ours

*a wooden treasure chest filled with gold coins and a lock, 3d, single object.*

LGM    LaRa   Meshy-3*   Luma Genie*   Ours

Figure 5: Qualitative comparisons between generated 3D assets and ones from related works. '*' refers to the non-publicly available commercial software.

roughness, depth, and normal map. We follow the experience from Long et al. (2024) to use the subset with LVIS annotations in the G-Objaverse dataset as our training set, which consists of 21,469 objects, and we name it as **LVIS subset**. Besides, to boost our model in generating correct PBR materials for Gaussian splats, we filter the whole dataset with the following criteria: (a) the object must have rendered albedo, metallic, and roughness; (b) the front, left, back, right, top, and bottom views have more than 20% meaningful pixels; (c) corresponding prompt has no specific words, including 'resembling', 'debris', and 'frame'. With the above criteria, we further select a subset consisting of 24,010 instances, which we refer to as **PBR subset**, and we fine-tune our model on this subset to improve the generation quality on PBR materials.

**Implementation details.** We implement our model based on MVDream 2.1 Shi et al. (2023b) and MV-normal-depth Qiu et al. (2024), the former shares the same structure with stable diffusion 2.1, and the later is the same as stable diffusion 1.5. Our model outputs four views of Gaussian attribute maps at the resolution of 128, which forms a dense 2DGS representation with 65,536 primitives. We first train our model on 8 RTX 3090 GPUs for approximately 17 hours on the **LVIS subset** and then fine-tune the model on 4 RTX 3090 GPUs for approximately 10 hours on the **PBR subset**.

### 4.2 QUANTITATIVE COMPARISON

**Quantitative comparison with multi-view diffusion model.** As our model is an adapter that adapts MVDream for direct 3D generation, we provide a quantitative evaluation as MVDream does. Specifically, we randomly choose 1,000 prompts from the training set and choose the rendered images as the target dataset. Then, we infer our model with the prompts and camera poses to generate the corresponding MV images and meshes, and we render 12 views around the generated meshes. The FID Heusel et al. (2017), Inception score(IS) Salimans et al. (2016), and CLIP score Radford et al. (2021) are reported in Table 1a.

|  | FID↓ | IS↑ | CLIP score(ViT-B/L)↑ |
|---|---|---|---|
| Training data (4 views) |  | 14.36 ± 0.69 | 33.43 |
| MVDream* | 32.06 | 13.68 ± 0.41 | 31.12 |
| MVDream | 35.42 | 13.83 ± 0.53 | 34.23 |
| Training data (12 views) |  | 14.27 ± 0.35 | 33.26 |
| Ours (12 views) | 33.02 | 14.13 ± 0.55 | 32.96 |

| Methods | Mesh Votes(%) | Appearance Votes(%) |
|---|---|---|
| LGM | 9.6 | 10.3 |
| LaRa | 30.0 | 14.4 |
| **Ours** | **60.4** | **75.4** |

(a) Quantitative evaluation of our methods, '*' means we cite the metrics from the original paper, as MV-Dream is evaluated on the non-public dataset; therefore, we report them here for reference only.

(b) User study against LGM and LaRa on the generated meshes.

Table 1: Evaluation of LISA model.

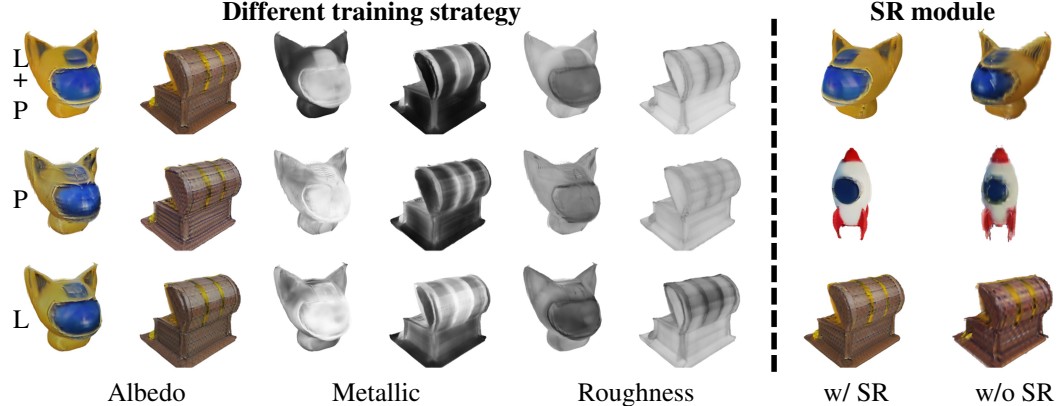

Figure 6: Ablation study on the super resolution (SR) module and training strategy. 'L+P' represents training on LVIS subset and fine-tuning on PBR subset, 'P' stands for PBR subset training only, and 'L' stands for LVIS subset training only.

**User study.** We further perform a user study to evaluate the performance of our pipeline. As our model is most related to Gaussian splats-based multi-views reconstruction methods, we perform comparisons with LGM and the SOTA method LaRa Chen et al. (2024a). We choose 150 generated results from Section 4.2, and reconstruct meshes using the generated MV images via the official scripts from LGM Tang et al. (2024) and LaRa Chen et al. (2024a). We render 360-degree circle views around the reconstructed meshes with and without texture maps, resulting in 300 videos. We present each volunteer with 20 multi-view images with corresponding prompts, then randomly show the rendered videos with or without texture maps from all methods, and ask everyone to vote for the best results to evaluate the appearance quality or geometry quality. We finally get 72 valid results and report them in Table 1b, which demonstrate the superiority over Gaussian splats-based 3D generation methods both in terms of appearance and geometry.

### 4.3 QUALITATIVE COMPARISON

As shown in Figure 5, we compare our results against other Gaussian splats based 3D generation methods, which indicate that our pipeline can robustly generate better meshes with decomposed PBR materials, while LGM and LaRa bake RGB into a single texture map. Besides, we also compare our results against Meshy-v3 mes and LumaAI-Genie Ai, which are non-publicly available software that supports 3D mesh generation with PBR materials. As in Figure 5, our model provides meaningful PBR material decomposition, and we further exhibits more results in Figure 7.

### 4.4 ABLATION STUDY

**Fine-tuning on PBR subset.** We train our model with the following different strategies: (a) **LVIS subset** training and **PBR subset** fine-tuning; (b) **PBR subset** training only; (c) **LVIS subset** training only. As shown in Figure 6, the LVIS subset helps model better reconstruct the geometry, and the PBR subset improves the PBR material generation.

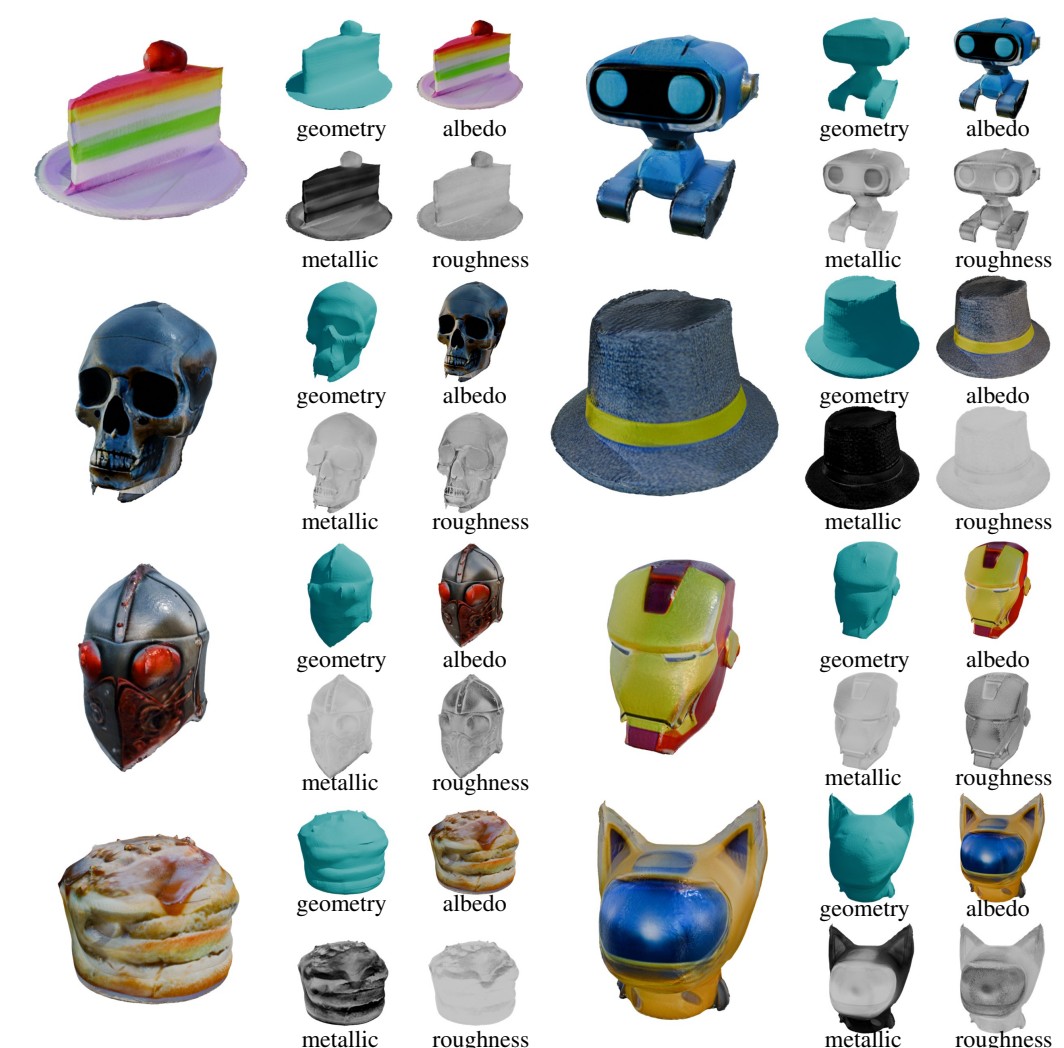

Figure 7: More text-to-3D results.

**Super resolution module.** We replace the upsampler module in our network with bicubic interpolation to upsample the feature maps to the resolution of 128, and train the model on **LVIS subset**. The results in Figure 6 shows that the super resolution module generates results with more details, while simply increasing the number of Gaussian primitive fails to improve the quality.

## 5 CONCLUSION

In this paper, we explore another way to exploit 2D diffusion priors for 3D generation except for socre distillation or generation-and-reconstruct scheme. We propose a novel LISA model that modulates and reassembles pre-trained layers from multi-view diffusion models for direct multi-view splat image generation, which demonstrates superior generalization ability and efficiency. Furthermore, we construct a complete pipeline based LISA model, which achieves efficient generation of high-quality 3D meshes with PBR materials on consumer-grade GPUs within 30 seconds. Quantitative and qualitative comparisons demonstrate that the great potential of our scheme.

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
