# OpenReview forum: "LISA: UNLEASHING 2D DIFFUSION FOR 3D GENERATION VIA LIGHTWEIGHT IMAGE SPLATS ADAPTATION"
_ICLR.cc/2025/Conference — ICLR 2025 Conference Withdrawn Submission_

### Official Review · Reviewer_JPgj · 2024-10-22

**Soundness:** 2
**Presentation:** 3
**Contribution:** 2
**Rating:** 3
**Confidence:** 5

**Summary:**

This paper focuses on text-to-3d generation with 2d diffusion prior. Previous methods often apply score distillation or adapt pretrained 2d diffusion model for multi-view generation. In contrast, this work proposed to adapt pretrained multi-view diffusion for direct generation of image splatting that has been used in LGM. However, this idea is not new, as [1] [2] and many other works have tried to adapt stable diffusion for direct triplane generation. There is a lack of discussion and comparisons with them. Also, the quality of generated results, including both geometry and PBR are not promising.

[1] PI3D: Efficient Text-to-3D Generation with Pseudo-Image Diffusion
[2] HexaGen3D: StableDiffusion is one step away from Fast and Diverse Text-to-3D Generation

**Strengths:**

- The idea of the paper is straight, and the writing is easy to follow.

**Weaknesses:**

- Discussion of novelty. The main claim of this paper is to adapt an image diffusion model for direct 3D generation. However, this idea is not new, as [1] [2] and many other works have tried to adapt stable diffusion for direct triplane generation. There is a lack of discussion and comparisons with them. The difference is 3D representation, where this paper uses an image splatter for Gaussian splattings. But this representation has been widely used in 3D generation, such as LGM, so this is not new, either.

[1] PI3D: Efficient Text-to-3D Generation with Pseudo-Image Diffusion

[2] HexaGen3D: StableDiffusion is one step away from Fast and Diverse Text-to-3D Generation

- Quality and evaluation. The quality of generated results, including both geometry and PBR, is not promising. For example, in Fig. 1, the albedos of both owl and panda are entangled with shadows, which are also presented in relit results. Also, the predicted metallic map contains metal parts in a wooden owl, which does not make sense. The generated geometry lacks details. These issues are common in shown results in the main paper. Also, all shown cases are very simple objects, which have been well addressed by existing methods. It also lack comparison with state-of-the-art methods like CLAY or rodin gen.

- Quantitative results. Why table 1 (a) only gives comparison with mvdream? How about FID and clip score of LGM and other SOTA methods? How about the metrics of the ablation study?

- Motivation and method. It should involve more analysis and insights of the proposed method. Why pretrained image diffusion model help the generation of splatter image? Gaussian splatting images are quite different from natural images in pretrained diffusion. so would the prior still help? I suggest comparing LISA  with a trained-from-scratch model and directly fine-tuning mode to examine whether the pertaining helps.

**Questions:**

- How about failure cases? How about the limitations?
- The proposed method involves a texture refinement process. How about the computing time? Is this important for the results? If yes, it needs some ablation study.
- The citation formats in this paper are wrong. All citations lack "()". It should use \citet or \citep instead of \cite.

---

### Official Review · Reviewer_8cnp · 2024-11-01

**Soundness:** 3
**Presentation:** 2
**Contribution:** 3
**Rating:** 5
**Confidence:** 4

**Summary:**

The paper presents an efficient framework for adapting a multi-view diffusion model to generate 3D assets represented through Gaussian Splatting. Unlike previous approaches, this method reuses feature maps from a pre-trained, frozen diffusion model, resulting in more efficient training and a lightweight model design. Additionally, the paper explores improved mesh extraction using 2D Gaussian Splatting (2DGS) and Physically-Based Rendering (PBR) materials, guided by multiple diffusion priors. Experimental results validate these claims, showcasing the method’s enhanced performance.

**Strengths:**

* The proposed lightweight 3D adaptor is both novel and effective, offering a new approach to leveraging 3D information within a multi-view 2D diffusion model.
* The method’s efficiency significantly reduces the compute requirements and training time.
* Extensive experiments are conducted to validate and support the proposed contributions.

**Weaknesses:**

* The methodology section lacks clarity, making it difficult to understand or reproduce. Since diffusion inference involves multiple steps, it remains unclear how the adaptor manages each step and how the final Gaussian Splatting (GS) is generated.
* Certain design choices are insufficiently explained or lack ablation studies. For example, why is only noise level under 80 used, and why are 16 groups of Gaussians selected?
* The paper contains several formatting and writing issues. For instance:
  * Many citations are missing parentheses (should use \citep{} instead of \cite{}).
  * Line 319: "gradient checkpointing" is a memory reduction method, I guess "gradient accumulation" may be what the authors intended.
  * Line 269: "Tough" should be corrected to "Though."

**Questions:**

* The process by which the model predicts the final Gaussian Splatting (GS) from the diffusion steps is unclear:
  * In Equation 1, it’s ambiguous what $i$ represents. Since there are multiple diffusion steps $t$ throughout the generation process, does $i$ correspond to different noise levels?
  * The rationale behind having 16 embedding vectors is also unclear. The paper mentions that only features from noise levels lower than 80 are used, does this imply that 16 diffusion steps are involved?
  * What distinguishes these 16 pairs of GS? Additionally, there is no explanation or ablation study on how this number was chosen.
* I’m also curious about the differences between the model in Figure 4 (trained on 2 RTX 3090 GPUs) and the final model (using 8 RTX 3090 GPUs). The initial model seems to perform well; how do extended training time and additional data contribute to improvements?

---

### Official Review · Reviewer_VZCR · 2024-11-03

**Soundness:** 2
**Presentation:** 2
**Contribution:** 1
**Rating:** 5
**Confidence:** 3

**Summary:**

This paper presents LISA (Lightweight Image Splats Adaptation), an approach to leverage 2D diffusion models for 3D asset generation. The key idea is to integrate a lightweight adapter into a frozen 2D diffusion model's decoder to simultaneously generate RGB images, Gaussian splats, and physics-based rendering material maps. The method enables direct feed-forward generation of relightable Gaussian splats, which can be converted into 3D meshes through inverse rendering.

**Strengths:**

- the method is leveraging a pre-trained 2D diffusion model as prior for its 3D generation task
- the integration of physics-based rendering material maps adds utility for graphics pipeline integration
- feed-forward generation could potentially enable faster 3D asset generation

**Weaknesses:**

- the novelty is incremental given the recent surge of papers combining 2D diffusion models with 3D generation
- limited discussion of failure cases and limitations
- the paper would benefit from more ablation studies on the adapter architecture choices

**Questions:**

- how sensitive is the method to the choice of the base 2D diffusion model?
- what are the limitations in terms of geometry complexity that can be handled?
- how the material maps can be reconstructed well, given the material-lighting ambiguities?
- how does the method handle multi-view consistency, especially for complex objects?

---

### Official Review · Reviewer_syqm · 2024-11-03

**Soundness:** 3
**Presentation:** 3
**Contribution:** 2
**Rating:** 5
**Confidence:** 4

**Summary:**

The paper proposed a method to adapt features from pre-trained 2D diffusion models to 3D asset represented by Gaussian Splatting.

**Strengths:**

1) The proposed method is light-weight and easy to implement.  As described in the paper, the training is done just on RTX 3090 GPU within 1 day.
2) The proposed method can be extended to in-cooperate other priors as described in Sec 3.3.
3) The experiments results demonstrate that the proposed method is effective in text-to-3d tasks, as well as the relighting tasks.

**Weaknesses:**

1) The authors claim that they exploit 2D diffusion prior for 3d generation. Specifically, it depends on a multi-view 2D diffusion model. The difficulties to use 2D diffusion model / multi-view 2D diffusion model are totally different. When a set of multi-view image is ready, the 3D asset is almost off-the-shelf. There are many works employ 2D multi-view model to generate 3D assets. For example, cat3d[1], mvssplat[2]. So why do we need to design a adapter to transform the multi-view diffusion features to 3D?

2) Following 1),  If the proposed method is superior or can be used to improve the base model of multi-view-to-3d methods, more comparisons are necessary. Currently, only experiments on MVSDream is provided.

3) Maybe more generated samples are helpful to prove the effectiveness of the proposed method. And the multi-view rgb images from the 2D diffusion model are needed for comparison.

4) Too many typos in writing. e.g. Figure 4(b) in line 283-284. where is Fig 4b? ; line 269, Tough-> Though;...

[1]Gao, Ruiqi, et al. "Cat3d: Create anything in 3d with multi-view diffusion models." arXiv preprint arXiv:2405.10314 (2024).

[2]Chen, Yuedong, et al. "Mvsplat: Efficient 3d gaussian splatting from sparse multi-view images." arXiv preprint arXiv:2403.14627 (2024).

**Questions:**

1) Any possibility to use single view 2D diffusion models as the base model? It should be a big step if it works, compared with the time-consuming SDS-based methods.
2)  The paper mentioned in line 296 "we leverage pre-trained a lightweight super-resolution model Wang et al. (2021b) to upsample the
feature maps". Does it make sense that the model pre-trained on natural images can be used to upsample the features in your work?
3) Please see the weaknesses I proposed before.

---

### Note · Authors · 2024-11-26

I have read and agree with the venue's withdrawal policy on behalf of myself and my co-authors.